# A Multi-scale Graph Network with Multi-head Attention for Histopathology Image Diagnosis

Submission ID 1135

No Institute Given

**Abstract.** Hematoxylin-eosin (H&E) staining plays an essential role in brain glioma diagnosis, but reading pathologic images and generating diagnostic reports can be a tedious and laborious work. Pathologists need to combine and navigate extremely large images with different scales and to quantify different aspects for subtyping. In this work, we propose an automatic diagnosis algorithm to identify cell types and severity of H&E slides, in order to classify five major subtypes of glioma from whole slide pathological images. The proposed method is featured by a pyramid graph structure and an attention-based multi-instance learning strategy. We claim that our method not only improve the classification accuracy by utilizing multi-scale information, but also help to identify high risk patches. We summarized patches from multiple resolutions into a graph structure. The nodes of the pyramid graph are feature vectors extracted from image patches, and these vectors are connected by their spatial adjacency. We then fed the graph into the proposed model with self-attention and graph convolutions. Here, we used a multi-head self-attention architecture, where same self-attention blocks are stacked in parallel. As proven in Transformer networks, multiple attention maps herein capture comprehensive activation patterns from different subspace representation. Using the proposed method, the results show a 71% accuracy for glioma subtyping. The multiresolution attention maps generated from the proposed method could help locate proliferations and necrosis in the whole pathologic slide.

**Keywords:** Whole slide image classification · Graph convolution network · Attention.

## 1 Introduction

Whole slide imaging (WSI), also known as virtual microscopy, scans a complete tissue sample and creates a single high-resolution digital image. During imaging, the scanner goes through the entire slide and captures high resolution image by patches and then stitches them into one large image. Herein, we refer whole slide images as images or slides and image patches as patches for convenience. The images could be visualized in various resolutions for pathological reading. Brain glioma images are usually scanned under a high magnification setting as 400x with 10x eyepieces and 40x objective lenses. We will use the objective lenses magnification in the following section. The task for pathological reading is to

identify abnormal areas and also to focus on detailed cell types. Based on cell types, brain glioma is classified into Oligodendroglioma (O) and Astrocytoma (A). Based on severity, Anaplastic Oligodendroglioma (AO) can be further identified from O, and subgroups Anaplastic Astrocytoma (AA) and Glioblastoma (GBM) of A can be classified. High risk glioma AO and GBM are generally identified by image appearance markers such as Micro-vascular Proliferation (MVP) and Necrosis (NEC) at a lower resolution [4].

Automatic recognition of WSI is necessary to assist pathologists, but the use of WSI in routine pathological workflows encounters challenges. First, WSI images contain thousands of high resolution patches, and it is difficult to train a deep learning model. Second, labeling image patches and identifying disease-related regions of interest (ROIs) can be laborious. Third, classification of glioma requires multiresolution assessment of images.

Considering these challenges, multi-instance learning (MIL) is naturally suitable for assisting WSI diagnosis. MIL assumes the object to be classified is a bag containing multiple instances (patches). Hashimoto et al. [2] proposed a domain adversarial training strategy for adding multi-scale information in MIL learning. Zhao et al. [8] used a graph structure to model the spatial correlation among different instances and introduced graph convolution layers to process such correlation. However, they are not suitable to handle both multiresolution images and spatial adjacency across resolutions at the same time. Moreover, extra labeling for ROIs is required, making preprocessing more complex.

Interpretability remains another challenge for MIL algorithms. Self-attention block has also shown great potential not only in the network explanation but also in feature representation. Transformers has raised lots of research interest in the NLP field [5]. It used a special multi-head attention block to cover the information from different hidden spaces at different positions. Although multi-head attention block is still an intuitive method and activation at different head are hard to merge, it still outperforms single-head attention block in many aspects, especially on multi-class classification tasks [7].

To address the aforementioned problems, in this paper we proposed a graph-based attention network to classify five types of glioma using pathological patches in multiresolution framework. The innovations of our proposed method is of three aspects: 1) We employed a multiple instance strategy where all eligible image patches are used and do not require any complex patch selection step; 2) We proposed to model the relations of image patches under different resolutions in a pyramid-like graph structure; 3) We proposed to use a multi-head graph attention pooling block to further cover the information from different subspaces jointly and improve the explainablity of our model.

## 2    Methods

We proposed the method below to classify whole slide images and also to identify ROIs under different resolution. We summarized each whole slide image as a graph, and introduced a graph convolutional based attention block to obtain

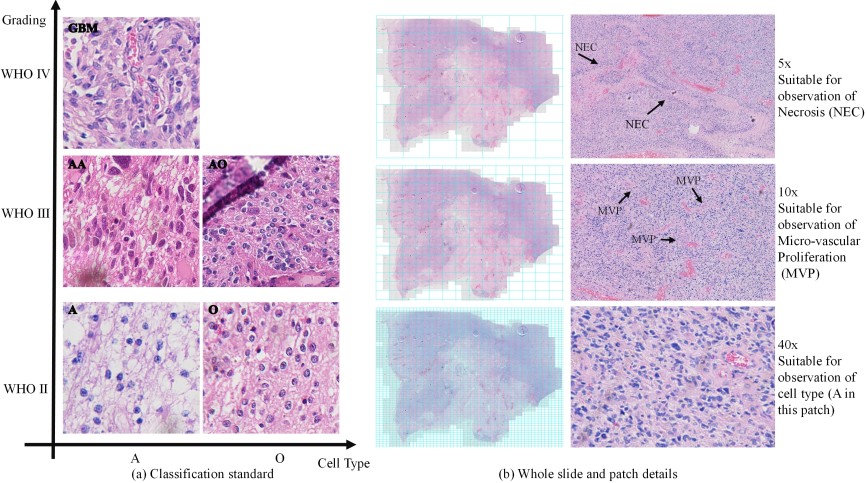

**Fig. 1.** WHO classification standard (a) and sample patches (b). Black arrows point at MVP and NEC regions, important image markers for glioma subtyping.

a prediction. First, we resample the whole slide image into three resolutions: 5x (best resolution for the detection of NEC), 10x (best resolution for the detection of MVP), and 40x (best resolution for observing cell type), as is shown in Fig. 1. Second, the pyramid-like histopathology graph is constructed by two components, the features of image patches extracted by a pre-trained DenseNet as nodes, and their spatial adjacency as edges. Patches under different resolutions are also connected if the high resolution image is corresponding to the low resolution image, as is shown in Fig. 2. Third, a multi-head graph pooling block is added to embed latent representations of the input graph and makes a prediction.

**Multi-resolution Graph Construction.** We resampled the whole slide images into several patches under different resolutions. In this case, we could obtain $N$ 5x image patches, $4N$ 10x image patches and $64N$ 40x image patches. We then introduce a multi-resolution structure to model the spatial relations of these patches. Image patches under the same resolution are connected if they are spatially adjacent. Image patches under different resolutions are connected if the high resolution patches are cropped from low resolution patches. After defining the graph structure, a M-dimensional feature vector extracted by a pre-trained DenseNet is assigned to each image patch. This multiresolution pyramid-like graph structure can be mathematically described by two matrices, a node feature matrix $X \in R^{69N \times M}$ and an adjacency matrix $A \in R^{69N \times 69N}$.

**Graph Attention Pooling.** We adopted a graph based self-attention opera-

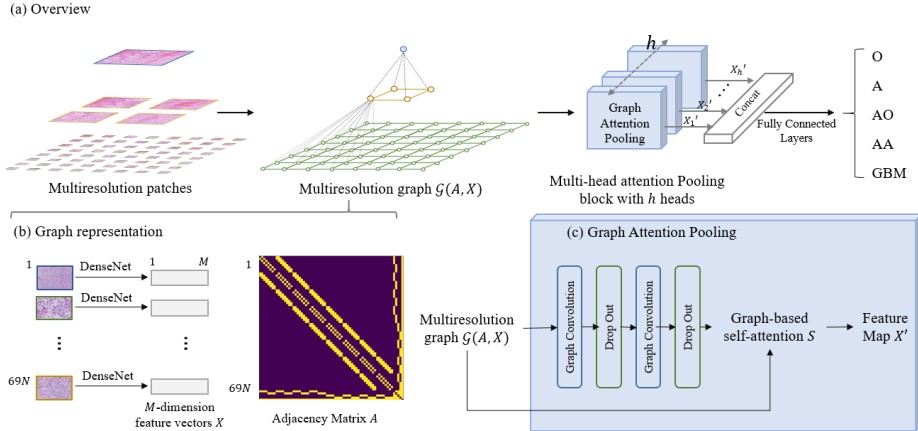

**Fig. 2.** The framework of the proposed Multi-resolution Graph Multi-head Attention Network (a), graph construction (b) and the graph attention pooling block. Adjacency matrix is defined based on spatial adjacency and cross-scale relationship. Here we used $h$ attention blocks to obtain comprehensive attention maps for five-class classification.

tion on every image patches. Traditionally, for an input feature map $X \in R^{N \times M}$ self-attention operation for the first dimension is defined as :

$$
\begin{aligned}
S &= (tanh(XW_1))W_2, \\
X' &= softmax(S^T)X.
\end{aligned}
\tag{1}
$$

Here, $W_1 \in R^{M \times p}$ and $W_2 \in R^{p \times 1}$ are weights of linear transformations while $p$ is the dimension of the hidden representations. $S \in R^{N \times 1}$ is the self-attention matrix whose entries are the attention score of every rows of $X$. $X' \in R^{1 \times M}$ represents the pooled vector. We further introduced a graph convolution based self-attention block to utilize the spatial adjacency between every image patches. The graph convolution operation of message passing model is defined as

$$
X^{l+1} = X^l *_{\mathcal{G}} W = ReLU(AX^lW).
\tag{2}
$$

For input feature vector at layer l, $*_G$ represents the graph convolutional operation and $W$ is the weight of graph convolutional transformation. $A$ is the adjacency matrix describes the connections between nodes. After one layer of graph convolution, the feature vector $X^{l+1}$ is obtained. Thus, by introducing graph convolutions to self-attention block, we could obtain graph attention pooling operations as below:

$$
\begin{aligned}
S_{\mathcal{G}} &= (tanh(X^T *_{\mathcal{G}} W_1)) *_{\mathcal{G}} W_2, \\
X' &= softmax(S_{\mathcal{G}}^T)X.
\end{aligned}
\tag{3}
$$

It is worth noting that, if omitting the numeric difference caused by activation functions, the proposed method can easily be transformed into traditional self-attention if we let $A$ be an identity matrix. Besides, in order to avoid value shifting during training, we normalized the adjacency matrix before graph attention pooling layer. The graph attention layer proposed here is different from Graph Attention Network (GAT) [6] because the purpose of the proposed method is to generate activations according to node features, instead of connections.

**Multi-head Attention.** Our goal is to classify five major types of glioma, and these glioma types are classified on different aspects of the H&E pathological images across different resolutions. Pathologists observe the color and texture of image patches, and identify the existence of specific biomarkers (MVP, NEC, and etc.). A 5-class classification problem arises activations at different positions in different subspaces. For example, the location of a typical O patches might be different from the locations of important grading markers. In this case, we added a multi-head attention scheme to obtain a comprehensive activation map of hidden spaces instead of a single attention function. For multi-head graph attention block with h heads,

$$X' = Concat[softmax(S_{(\mathcal{G},1)}^T)X, \ldots, softmax(S_{(\mathcal{G},h)}^T)X]. \tag{4}$$

Since we added dropouts and batch normalizations in the multi-head attention layer during training, our method could also be interpreted as the Bayesian training of attention maps. The variance of attention maps measures the uncertainty of activations.

## 3   Experiments

### 3.1   Dataset

We managed to acquire 440 whole slide images from an open source dataset [1]. We first proposed a simple but efficient RGB-value filter to remove background. Briefly, for images patch with RGB value difference among three channels are less than 7, this patch is removed from the following process.

Each whole slide image could compose 40-200 5x image patches, 100-700 10x image patches and 2000-7000 40x images. We further randomly split the 440 subjects into 200 training, 100 validating and 140 testing subjects. In total, there are 104 O patients, 104 A patients, 51 AO patients, 83 AA patients and 98 GBM patients.

We utilize a pre-trained classification model from a recent work [3]. In that work, a structure of DenseNet121 baseline with Squeeze and Excitation Blocks was used to classify Glioma subtypes. It used patches from a user-defined region (usually cover only 300 patches per slide), scanning from a different microscope with ours, and there are no intersections between our datasets and the datasets used in [3]. This model is used here only for feature extraction. Briefly, the image patches in this study are resized into $384 \times 512$ from $1824 \times 2720$ as inputs to

the model, and we used the feature maps from the last layer before the fully connected layer with the size of $M = 24 \times 32 = 768$ as features of every image patch.

## 3.2   Results

We computed classification accuracy, specificity and sensitivity with a number of methods and different settings of our proposed algorithm.

**Multi-head Graph Attention.** To verify the effectiveness of the proposed multi-head graph attention, we used the methods below for ablation studies and the results are shown in Table 1. The metric used is the recall of every class, which is calculated by (True Positive)/(True Positive+False Negative). We further measured the classification performance of cell type and WHO grade. The five major types can be summarized into two cell types, O (O, AO) and A (A, AA, GBM); and according to the WHO grading, into WHO II (A, O), WHO III (AA, AO) and WHO IV (GBM).

**Table 1.** Ablation study results. *Voting*: We used the pretrained model as predictor and assign a prediction by the labels of all image patches. *Max+Mean*: We replaced the graph attention pooling block by two pooling blocks, max pooling and mean pooling. And the results from two pooling blocks are concatenated together. *Multi-head Attention (h=4)*: We used a conventional multi-head attention block descripted in section 2. *Multi-head Graph Attention (h=4)*: This is our proposed method, which is featured by a graph-based multi-head self-attention pooling block.

| Methods | O | A | AO | AA | GBM | Cell Type | WHO Grade | Average |
|---|---|---|---|---|---|---|---|---|
| Voting | 0.09 | 0.20 | 0.68 | 0.78 | 0.88 | 0.81 | 0.58 | 0.50 |
| Max + Mean | 0.68 | 0.85 | 0.56 | 0.00 | 0.83 | 0.81 | 0.71 | 0.60 |
| Multi-head Att (h=4) | 0.46 | 0.85 | 0.56 | 0.50 | 0.90 | 0.82 | 0.80 | 0.67 |
| Multi-head Graph Att(h=4) | 0.81 | 0.61 | 0.5 | 0.75 | 0.77 | 0.80 | 0.84 | 0.70 |

**Attention Map Visualization.** We visualize the attention map from each attention head, as shown in Fig. 3. We derived the attention maps from model with h=8, and normalized them according to their head index. We averaged the activation maps, and selected the most activated patches in different resolutions.

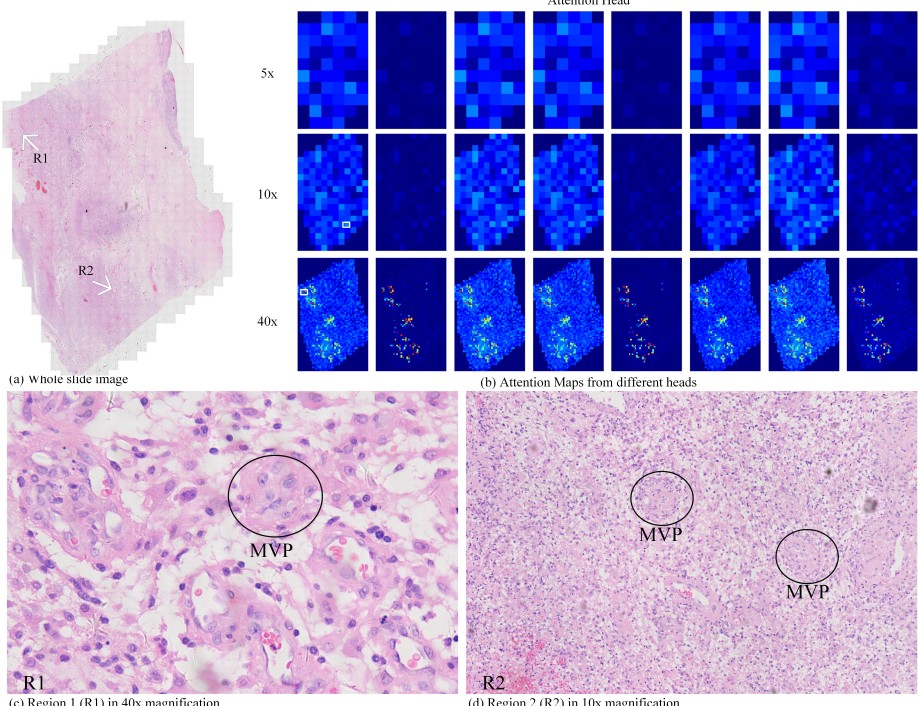

Attention Head

(a) Whole slide image

(b) Attention Maps from different heads

(c) Region 1 (R1) in 40x magnification

(d) Region 2 (R2) in 10x magnification

**Fig. 3.** Attention maps of a GBM case. Whole slide image (a), attention maps (b), cell shapes (c) and image markers (d).

Best visualization performance can be obtained at digital manuscript since the images are in high resolution.

We claim that 1) our attention maps in 40x magnification have shown a consistency with the cell distribution density, where the high density regions are activated; 2) our attention maps in 50x and 100x could help to locate the regions of NEC and MVP; 3) two major patterns of activations are shown in activations from different heads. It indicates the efficacy of multi-head attention block than that of single head attention. We also notice that mostly, activated patches appeared under 40x resolution.

**Feature Extractor Selection.** We also tested the proposed model on VGG-16 pretrained merely on ImageNet to evaluate the performance improvement brought by pretraining. For classifying A and O, the proposed method using VGG-16 extracted features achieves 84.2% accuracy, while the proposed method using pretrained network achieves 88.5% accuracy.

## 4   Conclusion

In this paper, we proposed a graph based multi-head attention network with multiple instance learning strategy for histopathological image diagnosis. We verified the proposed method of classifying five major subtypes of glioma. Our method is featured by a pyramid-like graph structure for input, a multiple instance learning strategy and a multi-head attention network for interpretation. Besides, since all image patches from different resolutions are used, our method do not require extra labeling and complex pre-processing steps. The graph-based multi-head attention block not only allows a simple permutation variant dimension reduction operation, but also covers activations at different positions for different feature subspaces. Our method has shown great potential in the diagnostic analysis of histopathological images. The attention map generated from our method could also help pathologists to identify and locate important image markers in whole slide images of high-risk gliomas. There are still some limitations of this work. For example, our feature extractor was pre-trained by another labeled dataset. Instead of using the feature extractor directly, domain transferring in the application should be considered.

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
