# OpenReview forum: "A Multi-scale Graph Network with Multi-head Attention for Histopathology Image Diagnosis"
_MICCAI.org/2021/Workshop/COMPAY — COMPAY 2021_

### Official Review · Reviewer_fGbH · 2021-08-07
**A multi-resolution graph is constructed from H&E images and a graph neural network with multi-headed attention is used for glioma subtyping**

**Rating:** 6
**Confidence:** 4

**Review:**

The use of multi-resolution graphs to incorporate spatial information is an attractive but the authors do not explain or explore how the multi-resolution nature of the graph affects the graph convolutions. Edges on the graph are now inherently directed as depending on the direction of traversal, the node is either increasing or decreasing in resolution.

The use of a multi-headed attention pooling makes sense, especially for a multi-task problem but the single head case is never benched marked against. Additionally, visualizations are done for Att(h=8) but results are provided for Att(h=4), it would be useful to make these consistent.

Overall the paper is well organized and the idea is interesting, Figure 2 in particular is very useful.

---

### Official Review · Reviewer_dDLs · 2021-08-16
**Good approach but a lot of details are missing**

**Rating:** 7
**Confidence:** 4

**Review:**

This paper presents an automated approach to classify subtypes of glioma from whole slide images (WSI). The proposed approach is based on a multi-scale graph structure with attention-based multiple instance learning. The study is performed on 440 WSIs from publicly available data. For visualization, multiple attention maps were captured to show activation patterns from different magnifications.

The paper presents an interesting approach, however, below are some major and minor comments
- Results are not properly explained, it is not clear why there is a huge disparity between the results from Multi-head Att (h=4) and Multi-headGraphAtt(h=4), especially for class O (0.46 and 0.81) and AA(0.5 and 0.75)
- Adding a confusion matrix would also help better explain the results
- In the abstract, it is written that the proposed model achieves  71% accuracy for glioma subtyping but in Table 1 the value is 0.7
- The attention maps from the model are derived with h=8 while the quantitative results are presented with h=4, it is difficult to understand the significance of different values for h
- The experiment performed for the Feature Extractor Selection needs more details, if the proposed model achieved 88.5% accuracy then why it is not listed in Table 1. If this experiment was performed with a different setting
- The attention maps from 5x and 10x seems least helpful, the reason for this should be properly explained
- In the Attention Map Visualization paragraph, it is not clear where the following sentence came from “our attention maps in 50x and 100x could help to locate the regions of NEC and MVP”
- The following sentence needs proper citation “Pathologists observe the color and texture of image patches, and identify the existence of specific biomarkers”
- The following sentence needs to be revised in Section 3.1, “scanning from a different microscope with ours”
- It is not clear what is 69 in the last line of the Multi-resolution Graph Construction section
- Proper formatting and grammatical error need to be addressed

---

### Decision · Program_Chairs · 2021-08-25

Accept